# Qualitative Experience of Self-Exclusion Programs: A Scoping Review

**DOI:** 10.3390/ijerph20053987

**Published:** 2023-02-23

**Authors:** Cyril Devault-Tousignant, Nicolas Lavoie, Mélissa Côté, Sophie Audette-Chapdelaine, Anne-Marie Auger, Anders Håkansson, Magaly Brodeur

**Affiliations:** 1Faculty of Medicine and Health Sciences, Sherbrooke University, Sherbrooke, QC J1H 5N4, Canada; 2Department of Foundations and Practices in Education, Faculty of Education Sciences, Laval University, Quebec, QC G1V 0A6, Canada; 3Clinical Addiction Research Unit, Faculty of Medicine, Lund University, Box 117, 221 00 Lund, Sweden; 4CHUS Research Center, Sherbrooke, QC J1H 5N4, Canada

**Keywords:** gambling, self-exclusion, responsible gambling, review, addiction

## Abstract

Gambling disorder is a major public health issue in many countries. It has been defined as a persistent, recurrent pattern of gambling and is associated with substantial distress or impairment, lower quality of life, and living with a plurality of psychiatric problems. Many people suffering from gambling disorder seek help in ways other than formal treatment seeking, including self-management strategies. One example of responsible gambling tools that has gained popularity in recent years is self-exclusion programs. Self-exclusion entails individuals barring themselves from a gambling venue or a virtual platform. The aim of this scoping review is to summarize the literature on this topic and to explore participants’ perceptions and experiences with self-exclusion. An electronic literature search was conducted on 16th May 2022 in the following databases: Academic Search Complete, CINAHL Plus with Full Text, Education Source, ERIC, MEDLINE with Full Text, APA PsycArticles, Psychology and Behavioral Sciences Collection, APA PsychInfo, Social Work Abstracts, and SocINDEX. The search yielded a total of 236 articles, of which 109 remained after duplicates were removed. After full-text reading, six articles were included in this review. The available literature shows that although there are many barriers and limitations to the current self-exclusion programs, self-exclusion is generally viewed as an effective responsible gambling strategy. There is a clear need to improve the current programs by increasing awareness, publicity, availability, staff training, off-site venue exclusion, and technology-assisted monitoring, as well as by adopting more holistic management approaches to gambling disorders in general.

## 1. Introduction

Gambling disorder (GD) is a major public health issue in many countries [1,2]. According to the DSM-5, GD is defined as a persistent, recurrent pattern of gambling that is associated with substantial distress or impairment [3]. Prevalence studies across different countries in the world indicate that the lifetime prevalence of GD varies from 0.12% to 5.86% [2]. In addition to having a lower quality of life [4], 64% of individuals living with GD have been estimated to suffer from three or more psychiatric disorders [5]. Furthermore, only 10% of people with GD seek treatment, as demonstrated by an epidemiological study conducted in the United States [6]. However, it has been highlighted that many patients with GD seek help in ways other than traditional and formal treatment seeking, such as through different types of self-management strategies [7,8].

Responsible gambling initiatives aim to reduce gambling-related harm by limiting the gambling activities of affected gamblers [9]. One example of a responsible gambling program that has gained popularity in recent years is self-exclusion programs. Self-exclusion entails individuals barring themselves from a gambling venue (e.g., casino) or virtual platform [10]. For instance, exclusions can be requested by the person who gambles by formally signing an agreement, denying them access to a specific gambling venue, or to multiple ones [11]. Self-exclusion measures vary by jurisdiction. For example, in some jurisdictions, people who gamble and do not respect their self-exclusion programs are susceptible to punitive measures, such as fines or trespassing charges [12]. In New Zealand, for example, a breach of a self-exclusion order can lead to being asked to leave the premises by the venue staff, police officers could be called and a fine of NZD 500 could also be issued [13]. Although self-exclusion services typically include local, venue-specific, or operator-specific gambling, more uncommonly, other exclusion services involve all licensed gambling in an entire jurisdiction [14,15]. Self-exclusion programs can be implemented when gamblers feel that they cannot control their gambling, or when gambling interferes with their emotional, cognitive, or social capabilities [9]. Self-exclusion can also be used as a prevention tool, but seems to be under-utilized [10].

Different quantitative studies have tried to evaluate the effectiveness of self-exclusion programs, for example by using randomized trials [9,10,11,16,17,18,19]. Controlled trials are difficult to implement, which reduces the quality of the findings [11,20]. Many studies point to the following promising outcomes: gambling reduction, reduction in problematic gambling, improvement of aspects of the mental health of participants, and reduced money and time spent [9,10,20]. However, few studies have identified the key success factors for self-exclusion programs. In this context, it is crucial to understand the qualitative experience of gamblers who have used self-exclusion tools and programs to improve their effectiveness in the future.

According to Zhang [21], there are gaps that must be bridged to better understand the motivations, features, and challenges gamblers face when using responsible gambling techniques, such as self-exclusion. Given that people have different gambling motivations, that their experiences vary in complexity, and that they are affected in different ways, studies that focus on participants’ perspectives are needed to identify the most common issues facing participants who require self-exclusion.

This scoping review focuses on the experiences of the actors involved in the self-exclusion process and includes qualitative studies published in peer-reviewed journals. Thus, the purpose of this scoping review is to summarize the literature to explore participants’ perceptions and experiences with self-exclusion programs.

## 2. Materials and Methods

This study is a scoping review based on the methodological framework established by Arksey and O’Malley [22] and refined by Levac et al. [23], and follows the PRISMA guidelines [24]. Scoping reviews are used in emerging areas of research to provide an overview of the current state of the literature [23,25]. Scoping reviews aim to systematically map the existing empirical data, synthesize key concepts, and detect any potential gaps in knowledge. The five steps to conduct a scoping review are as follows: (1) identifying the research question; (2) identifying relevant studies; (3) selecting studies; (4) charting the data; and (5) collating, summarizing, and reporting results [22,23]. The main question guiding this review is as follows: “What is the current knowledge on the qualitative experience of self-excluders and other actors closely involved in self-exclusion programs?”

### 2.1. Data Sources and Literature Research

As illustrated in Figure 1, an electronic literature search was performed on May 16th 2022, through EBSCOhost Research Databases in the following 10 databases: Academic Search Complete, CINAHL Plus with Full Text, Education Source, ERIC, MEDLINE with Full Text, APA PsycArticles, Psychology and Behavioral Sciences Collection, APA PsychInfo, Social Work Abstracts, and SocINDEX. An experienced information specialist helped to develop a search strategy using a controlled vocabulary (MeSH) and keywords related to the concepts “self-exclusion” and “gambling”. For self-exclusion, we searched using the following terms: self-exclu* and “self exclu*”. For gambling, we searched using the following terms: gambl*, betting*, “electronic gaming machine*”, lotto, lotter*, casino, poker, bingo, blackjack, and “slot machine”.

### 2.2. Study Selection and Data Extraction

To be eligible for this review, a study had to meet the following inclusion criteria: (1) the primary theme of the study is directly about gambling self-exclusion, (2) the study uses a qualitative design, (3) the study is published in a peer-reviewed journal, and (4) the study is available in English. The search yielded 236 articles, of which 109 remained after duplicates were removed. Following the title and abstract review, 24 articles were retained. Six independent researchers—CD-T, NL, SA-C, MC, AMA and MB—reviewed each of the twenty-four articles retained for full-text reading, and discrepancies were resolved by consensus through detailed evaluation and discussion by the team members. Following the full-text review, six articles were retained. All papers citing the six selected articles were then analyzed to find works that would fit the inclusion criteria. No additional articles were found, so six papers were included in the review. Figure 2 illustrates this search strategy and study selection process. 

Two team members, CD-T and NL, extracted information from the six articles. SA-C and AMA, in collaboration with MB, supervised and validated this extraction. Descriptive data from these selected articles, such as the name of the authors, year of publication, country, objective, methodology, study population, and conclusion, were collected for each article (see Table 1). The results were summarized by CD-T and NL, under the supervision and validation of SA-C, AMA and MB. Narrative synthesis was used to report and synthesize the findings [26]. SA-C, MC, AMA, AH, and MB revised the synthesis.

## 3. Results

### 3.1. Study Characteristics

All six articles were published between 2012 and 2022. Five of the articles are from Australia [16,27,28,30,31], and one is from China [32]. All articles were original research articles, and they sampled study groups from the country’s population to assess the features, efficiency, and challenges of self-exclusion as a responsible gambling technique.

### 3.2. Methodology

For this scoping review, all articles selected used a qualitative methodology. Three studies used telephone interviews [16,27,28]. Tong et al. [32] and Pickering et al. [30] used focus groups to gather information about gambling behaviors. Lastly, Pickering et al. [31] conducted interviews using different methods (in person, telephone, and Skype).

### 3.3. Population Studied

Two of the selected studies included only participants who had self-excluded and who were recruited by the organization responsible for self-exclusion [27,31]. In Hing and Nuske’s work [27], participants were selected from the Independent Gambling Authority, a self-barring program operated by the South Australian government. In the case of Pickering et al.’s work [31], the authors included former and current participants in a multi-venue self-exclusion program in New South Wales, Australia. Hing et al. (2014) [16] chose to study participants who had experienced a gambling problem, with about half of their population having self-excluded and the other half having not used this strategy. Similarly, Hing et al. (2015) [28] chose to include internet gamblers who were either moderate-risk or problem gamblers. Of note, the participants in this study were only men, whereas the five other studies included women and men in similar proportions. In the case of Tong et al.’s work [32], the researchers decided to study casino employees and non-casino employees who were not problem gamblers. Similarly, Pickering et al. [30] included participants who were self-exclusion users, but also professionals directly involved with problem gambling, such as gambling counselors, venue staff, and policy makers.

### 3.4. Motivations and Initiation of the Self-Exclusion Process

Two studies found that the main goal of self-excluders was to completely stop gambling [16,27]. However, Hing et al. (2015) [28] noted that most people who gamble excluded themselves to stay within their limits.

According to the excluders, the most important motivation for self-exclusion was related to themselves as problem gamblers [16,27,28,31]. Gamblers need to recognize their problems before they begin the self-exclusion process, and this recognition remains an important factor throughout their journey [10]. Tong et al. [32] noted that many casino employees indicated that people who gamble had critical responsibilities for self-exclusion to work. As mentioned by a casino employee, “If the gamblers are not willing to apply for self-exclusion, no one can really force them to do so.” Hing et al. (2014) [16] found that many problem gamblers were not conscious of the severity of their gambling addictive behaviors, and thus did not feel the need to be helped. According to other studies, the decision to self-exclude involves other significant contributors, including family members, partners, friends, and counselors [16,27,31]. Two studies identified that financial, emotional, and relationship problems were important triggers for self-exclusion, with financial crises playing the largest role [16,27]. In fact, Hing and Nuske [27] included non-excluders in their study, and many indicated that they would only consider self-exclusion in the case of severe financial or relationship issues. They also noted that there was often a specific event that led to self-exclusion [27].

Although gamblers’ motivation was described as primordial to self-exclusion, casino employees were pessimistic regarding gamblers being capable of initiating self-exclusion on their own. Many of them pointed out that people who gamble often do not recognize themselves as having a problem, which in turn impedes the initiation process [16,32].

### 3.5. Perceptions of Self-Exclusion Programs

#### 3.5.1. Type of Gambling Activity

Self-exclusion can take a very different form according to the setting of the gambler. Whereas gamblers who self-exclude from land-based venues authorize staff to remove them from the gambling site if detected [9], online gamblers risk having their accounts suspended for a fixed amount of time. Although only one study gathered empirical data on online gambling [28], it is interesting to put the experience of people who gamble online into perspective with land-based gamblers, as there is a high prevalence of gambling-related harms in online gambling [35]. In general, people who gamble online believe that this specific type of gambling requires more intense responsible gambling measures than on-land gambling because of its distinctive characteristics. For example, Hing et al. (2015) [28] noted that many online gamblers knew that they were at a higher risk of losing control. However, they also emphasized that gambling operators are not proactive enough and that some platforms do not have any responsible gambling initiatives.

Overall, self-exclusion, when available, was perceived to be effective in reducing the internet gambling of moderate-risk gamblers and problem gamblers [28]. Due to the nature of online gambling, gamblers were more likely to use informal strategies, such as their own willpower, budgets for a particular timeframe, or restricting the amount used to gamble [28]. However, the participants were conscious that these strategies had very limited usefulness, especially when under the influence of alcohol or exposed to heavy advertising [28].

#### 3.5.2. Low Awareness and Low Publicity

An important barrier in the process was the lack of knowledge regarding self-exclusion programs, demonstrated by the following statement: “A lot of people don’t even know that they can self-exclude” [16]. Most interviewees highlighted that there was low publicity and very limited public information available for self-exclusion programs [16,27,28,31,32]. Self-excluders in Hing and Nuske’s [27] study complained about the absence of advertisements for self-exclusion programs, as none of them had heard of self-exclusion before they started acknowledging their gambling problem. Participants viewed the active promotion of the unique benefits of self-exclusion programs as essential to increasing gamblers’ uptake [30].

Venue staff and gambling operators were perceived by participants to have the responsibility of identifying at-risk gamblers and intervening when necessary [28,31]. In fact, most studies have found that counseling agencies play a critical role in educating problem gamblers regarding self-exclusion, as well as in referring them to self-exclusion programs, supporting them in the process, and even helping them arrange exclusion [16,27,30]. However, Hing and Nuske [27] found that very few excluders were directly targeted by venue staff to discuss their gambling issues. Even casino workers declared that the advertisements were insufficient and ineffective. In fact, casino workers also have limited insight into strategies to improve awareness [32].

Regarding online gambling, the gamblers highlighted the need for greater involvement of gambling operators. However, as noted by many participants, some gambling websites lack effective responsible gambling tools, such as self-exclusion. In addition to advocating for more advertisements for responsible gambling practices, people who gamble online pointed out the need to restrict gambling advertisements [28]. Lastly, in one study [16], some of the non-excluders mentioned that they were aware of self-exclusion programs, although the vast majority only had very superficial knowledge of them. Overall, there is a lack of awareness and effective advertising for self-exclusion programs.

#### 3.5.3. Privacy and Confidentiality

The participants of three studies identified confidentiality and privacy as major deterrents to self-exclusion [16,27,32]. Many excluders, non-excluders, casino employees, and non-casino employees pointed out that many problem gamblers are worried about information leakage. Problem gamblers may fear any disclosure to their relatives, friends, and colleagues, as this may cause adverse consequences, as well as social rejection [16,27,32]. In fact, excluders in Hing et al.’s (2014) study [16] were especially concerned with privacy and confidentiality during the registration process. Furthermore, confidentiality and privacy have been described as especially important in close-knit communities, as information can circulate easily [16].

In Pickering et al.’s (2022) study [30], the participants suggested that online registration of self-exclusion programs could solve these problems. The online platform would allow gamblers to bypass the need for an on-site visit to initiate the process, thus eliminating the risk of being caught by people they know. Participants all mentioned the importance of highly secure online data management measures to protect users’ confidentiality, but wished that the site stayed as user-friendly as possible [30].

#### 3.5.4. Registration Process and Recommendations

Most participants complained about the time-consuming and complicated process of self-exclusion programs [16,27,31]. An important recurring factor discussed was the ability to self-exclude from more than one gambling facility at a time (multiple venues exclusion) rather than having to complete the process for each location where gamblers wanted to exclude themselves (single venue exclusion). Excluders in Hing et al.’s (2014) [16] study mentioned that for the Queensland self-exclusion program, most facilities operate a single venue self-exclusion. Therefore, if self-excluders wanted to be barred from multiple venues, they had to go through the process several times, which was time-consuming. In the same study, many excluders were confident that multiple venue exclusion, in addition to reducing process time, would reduce the demand for a variety of resources, including time and money [16]. Furthermore, they argued that this would also increase efficiency because many individuals who gamble failed to self-exclude from several venues, and this would reduce compliance as they would gamble on sites. Shame and embarrassment were also concerns when visiting multiple venues for self-exclusion. Moreover, non-excluders also identified the time, transportation, and money needed to complete the process of exclusion as significant barriers [16].

By contrast, Pickering et al. [31] and Hing and Nuske [27] studied registration processes that allowed for multi-venue exclusion. Although most excluders in both studies supported the exclusion of multiple venues, many complained that the maximum limit of venues was too low [27,31]. As a result, a significant proportion of problem gamblers went to venues they were not barred from, demonstrated by the following statement: “I still am at risk if I travel to other places” [27].

As mentioned above, an online registration process can solve some of these difficulties [30]. Such a website could be implemented nationwide, thus eliminating the need to register more than once. Participants highlighted the importance of the registration process not being too time-consuming and believed that relevant online help resources should be present [30].

#### 3.5.5. Venue Monitoring for Breaches of Self-Barring Orders

In all four articles that studied on-land self-exclusion, many participants lacked confidence that venues effectively monitored breaches of self-barring orders. This was confirmed by the fact that some excluders did gamble at barred venues without being detected [16,27,31,32]. Reasons for poor venue monitoring included recognition difficulties, high staff turnover, large density of casinos, lack of motivation of some casinos to follow policies, superficial and inadequate training of employees, and conflicts of interest [16,31,32]. In other words, these articles highlighted the need for better detection systems.

#### 3.5.6. Attitude and Competence of the Staff

Gamblers’ perceptions of staff attitudes, behaviors, and helpfulness in the self-exclusion process were very heterogeneous across studies [16,27,28,30,31,32]. Even though Hing et al. (2014) [16] noted that some self-excluders in their study commented on the high availability, respectfulness, and customer-centric staff, there was some distrust of their competence by other participants. Casino employees were also perceived by gamblers as lacking sufficient training [31]. In addition to the importance of staff in monitoring breaches of self-exclusion orders, gamblers believed that they had to play a proactive role in informing problem gamblers of the availability of self-exclusion programs and intervening when necessary; however, the participants did not feel that the staff accomplished this [28,31]. Gamblers also felt that venue staff were sometimes insensible and even unhelpful [30]. Therefore, improving staff training to support, interact with, and detect problem gamblers is necessary to increase self-excluders’ satisfaction. In that sense, having an online self-exclusion platform was also perceived as useful by the staff, as this could provide important information, such as previous breach reports and relevant data collection [30]. Finally, the staff and online operators were perceived to be key in increasing gamblers’ awareness of self-exclusion programs and facilitating referrals to gambling counseling services [16,31].

#### 3.5.7. Perceived Outcomes

Overall, excluders perceived self-exclusion to be beneficial. There was some encouraging reduction in gambling rates among the respondents, which supports the perceived effectiveness of self-exclusion programs [16,27,28,31]. For example, according to Hing and Nuske [27], 85% of self-excluders ceased or lessened their gambling. Some participants highlighted other benefits, such as financial and emotional stability [27]. By contrast, non-excluders stated that although self-exclusion might be useful for some, they believed that it was just not for them [16]. Tong et al. [32] found that some casino employees believed the issue was with the individuals who gambled; therefore, they doubted the effectiveness of self-exclusion and made less effort to promote self-exclusion. In this study, self-exclusion was felt to be implemented very passively by the casinos [32].

## 4. Discussion

This scoping review focused on the experiences of the actors involved in self-exclusion processes. The objective was to summarize the literature to explore participants’ perceptions and experiences with self-exclusion programs. As raised by the participants of the studies, there are multiple barriers and limitations to the current self-exclusion programs, which are as follows: low publicity, low awareness, lack of privacy and confidentiality, insufficient staff training, inadequate monitoring of breaches, resources required for registration processes, and perceived ineffectiveness from some respondents [16,27,28,30,31,32].

As outlined in a study from the United Kingdom, less than 1 out of 5 people who gamble were aware of self-exclusion programs [36]. Canadian data from 2005 show that only 0.6% to 7.0% of problem gamblers signed up for self-exclusion [37]. More recent data show similar results [6,38], with the proportion of individuals with GD seeking formal help still under 10%. As it is a major cause of under-utilization of these programs, improving awareness among people who gamble is essential. Multiple recommendations were offered to improve the low publicity and awareness of self-exclusion programs. These recommendations included more upstream strategies, such as greater government action to fund responsible gambling strategies and the creation of preventative measures, including education in schools and through mass media marketing [32]. Moreover, gambling venues play an important role. Adequately trained staff and gambling operators providing supportive environments to increase awareness of self-exclusion programs using pamphlets, e-guides, information packets, email, guidance, and direct interaction with problem gamblers is essential. As described by some participants in Hing et al.’s study [28], there is more promotion of gambling than promotion of responsible gambling strategies.

Although not emphasized to a large extent in this review, another important barrier that is intrinsic to any social activity is the social benefits in the form of entertainment that people who gamble can experience [39]. Therefore, there should be more initiatives to educate the public about the cost-benefit ratio of gambling, as it can become a substantial source of harm [40].

In the six articles included in this review, there was great variability in the feedback of the participants regarding the self-exclusion process. Participants who gave the most positive feedback participated in the study by Hing and Nuske [27], who examined the self-exclusion program operated by a South Australian government regulatory body known as the Independent Gambling Authority. This program has multiple distinctive features. Notably, the Independent Gambling Authority is a centralized service that allows self-barring from multiple venues at once, and it allows doing so away from gambling venues. This centralized service organization seems to be advantageous over programs that require independent applications to self-exclude from multiple individual venues, as stated in previous research [10]. Furthermore, off-site multi-venue exclusion seems to have other advantages, including lower resource consumption and a reduced desire to gamble when compared to on-site exclusion processes. Moreover, as noted by participants in Hing et al.’s [16] and Pickering et al.’s [30] studies, off-site exclusion may help reduce stigma, embarrassment, privacy, and confidentiality barriers. The off-site exclusion process could be carried out via the internet, off-site venues, hotlines, and other indirect channels. Although it has multiple limitations, such as impulsive exclusions and technical complexities, most participants in Pickering et al.’s 2019 study [31] supported online self-registration to improve the effectiveness and privacy of the self-exclusion process. In a later study, Pickering et al. [30] explored the different issues related to the creation of such a platform, describing the ideal website attributes and possible obstacles, such as collecting and verifying personal information. Increasing the maximum limit of self-exclusion venues in the self-exclusion contract is also needed, according to most respondents in all articles included in this review. In this context, it may be argued that for online gambling types, self-exclusion, even when involving many operators, may be limited by the risk of breaching the exclusion through offshore gambling operators through websites that are easily accessed online [41].

As mentioned by many participants, counseling plays an important role in the self-exclusion process. In fact, another key feature of the Independent Gambling Authority is that people with problematic gambling habits are assisted by trained counselors; therefore, this allows for better collaboration to ensure counseling support. This is important because problem gamblers often have multiple psychiatric comorbidities [5] and counselors have been shown to be important for the decision to self-exclude [11]. Although most respondents in all of the articles were satisfied or believed in the effectiveness of self-exclusion, many casinos’ employees thought that counselors or social workers may be needed to improve the effectiveness of programs [32]. Participants in the study by Pickering et al. [31] emphasized the need for additional help through counseling, telephone, or internet support services, as they would be the best adjunct to self-exclusion to manage underlying psychological issues that lead to gambling addictions in the first place.

As mentioned before, gamblers’ intrinsic motivations seem to play the most crucial role in the self-exclusion process. However, there is evidence that families and counselors also act as important external support for the gambler’s motivation. Moreover, there was a strong emphasis on the need for well-trained, non-judgmental, respectful, and readily available staff. Overall, the quality and quantity of interactions with the staff and gambling operators were mixed. In other words, there was clear room for improvement in these aspects. As described by Hing et al. [16], among other studies, respondents felt that there was a need for venue staff to be more proactive in the self-exclusion process, for example, with links for referral to counseling. In fact, some participants in Tong et al.’s study [32] proposed in-venue counseling services that would provide instant support in the case of significant acute gambling-related harm, which would require the presence of on-site trained counselors. Furthermore, land-based studies have shown that there must be improvements in the monitoring of breaches [16,27,31,32]. This is consistent with the literature. For example, a Canadian study found that only 48% of people who gambled and who breached their agreement were recognized, and 81% of them stated that it was very easy to do so [42]. In addition to reminding staff to be attentive to the detection of self-excluders, artificial intelligence or technology-based monitoring could be useful to improve the detection of breaches in self-exclusion orders. In fact, technological systems, such as computerized and mobile applications, could enhance the speed and accuracy of detecting defaulters and enforcing punitive measures such as fines. While this is an innovative avenue, the implementation of these technologies raises important ethical issues.

Although online gambling sites benefit from easier monitoring of gambling behaviors, permitting the identification of markers such as increased staking behaviors, increased net expenditure, high variability, and frequency of gambling activity [43], responsible gambling tools, such as self-exclusion, tend to be less available for people who gamble online [28]. Furthermore, the risk of breaching a self-exclusion may be particularly high for online gambling, where access to overseas web pages online leads to a risk of gambling relapse, despite self-exclusion within the gambler’s own geographical setting. As electronic gaming machines tend to be associated with the greatest risk of harm out of all different types of gambling [44], this is a major issue when such high-risk gambling happens online. As mentioned in the section above, the effect of the deficiency in formal responsible gambling programs is that online gamblers more often use informal strategies, such as setting their own limits. However, there is evidence that informal strategies tend to be much less effective, especially for severe problem gamblers [28]. As an online self-exclusion platform, such as the one imagined by Pickering et al. [30], would, of course, include online gamblers, making this dimension of gambling available would also be a great option.

Finally, although there is evidence that intrinsic motivation is critical to the adoption of responsible gambling strategies, gamblers’ families’ distress and disapproval can also act as motivations for behavior modification [45]. The Family Exclusion Order, another alternative to self-exclusion, has also been perceived as an effective intervention [46]. The Family Exclusion Order is a third-party exclusion model in which relatives can obtain barring orders for family members that cause harm due to gambling behaviors or addiction [46]. Family Exclusion Order has been shown to reduce familial conflicts, as well as to lower gambling. However, similar barriers and limitations apply. The major limiting factor described by Goh et al. [46] is the lack of multi-venue exclusion programs, which lead to continued gambling in other venues. Even though self-exclusion may seem to better an individual’s volition, family exclusion can also help addicted individuals increase intrinsic motivation and nurture familial relationships [46].

## 5. Conclusions

This paper summarizes the qualitative literature on the experiences and perceptions of self-excluders, non-self-excluders, and other actors in the process of self-exclusion programs. Although there are many barriers and limitations to the current self-exclusion programs included in this review, self-exclusion is generally viewed as an effective responsible gambling strategy. There is a clear need to improve the current programs by increasing awareness, publicity, availability, staff training, off-site venue exclusion, and technology-assisted monitoring. Moreover, research must be conducted on the perceptions of all stakeholders of self-exclusion programs and their alternatives to optimize these programs. The integration of effective self-exclusion programs within a holistic management approach, in combination with counseling and other additional aid, will most likely have the most beneficial impact on the quality of life of problem gamblers, their families, their friends, their communities, and their societies.

## 6. Study Limitations

Some limitations to this review should be outlined. First, it contains only published peer-reviewed qualitative research articles; therefore, it does not include gray literature or quantitative studies. Moreover, only English-language articles were included. The gray literature articles, as well as studies in other languages, would have offered further insight into the subject. Furthermore, the review included five studies from Australia and one from China, which does not allow us to have the perspective of actors from other countries. Finally, multiple articles were published by the same authors, reducing our ability to include different perspectives or exclude biases.

## Figures and Tables

**Figure 1 ijerph-20-03987-f001:**
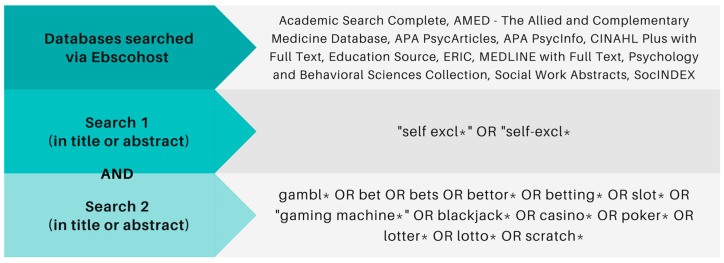
Keyword string of the search strategy (search date: 16 May 2022).

**Figure 2 ijerph-20-03987-f002:**
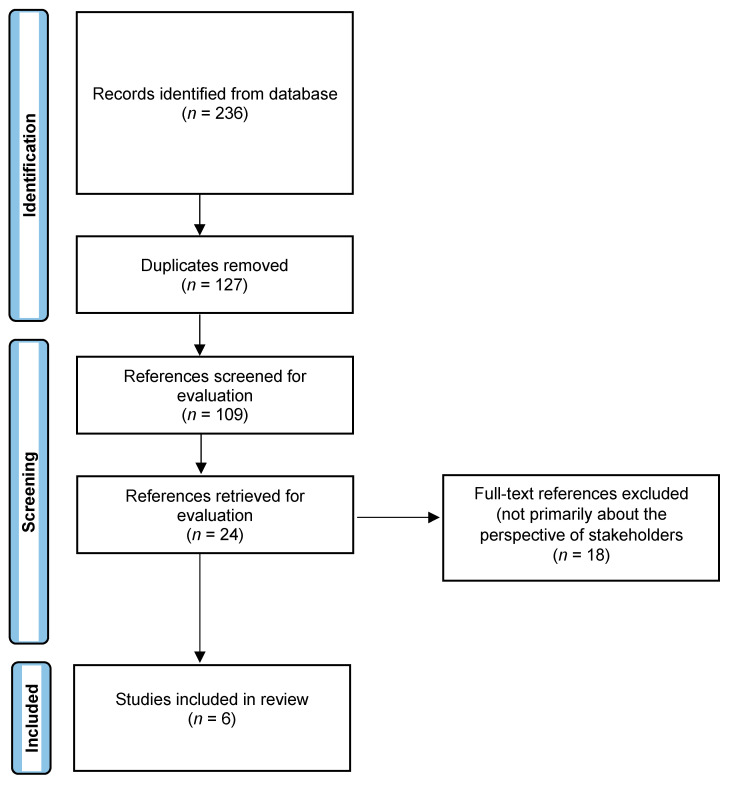
PRISMA flow chart of the search strategy and study selection.

**Table 1 ijerph-20-03987-t001:** Description of the included studies.

Author (Year)	Title	Country	Design	Objective	Methodology	Population	Conclusion
Hing and Nuske (2012) [27]	The self-exclusion experience for problem gamblers in South Australia	Australia	Original research	“While self-exclusion programs are widely available, little research has been conducted into their operations and efficacy, particularly from the self-excluders’ perspective. This paper presents findings from 36 survey responses and 23 interviews with gamblers who had self-excluded through a centralized service in South Australia.”	Qualitative research. Ethics approval and informed consent obtained. No instrument mentioned for scoring gambling problems	Phase 1 survey: *n* = 36 33% men 67% women Phase 2 telephone interview: *n* = 23 35% men 65% women Mean age = 46.1 Only self-excluders	“They identified key program shortcomings as low publicity, limits on how many venues they could self-bar from, and inadequate venue monitoring for breaches of self-barring orders. Nevertheless, the centralized service, staffed by trained psychologists and located away from gaming venues, which allows multiple venue barring in one application, appeared advantageous over programs that require people to self-exclude directly from individual gaming venues. Most respondents (85%) had ceased or lessened their gambling in the 12 months following self-barring. Nevertheless, some continued to struggle to manage their gambling, reflected in breaches of their orders and gambling in venues from which they were not excluded.”
Hing et al. (2014) [16]	A Process Evaluation of a Self-Exclusion Program: A Qualitative Investigation from the Perspective of Excluders and Non-Excluders	Australia	Original research	“This paper draws on a process evaluation of Queensland’ self-exclusion program to examine how people use the program, motivations for self-excluding, barriers to use, experiences and perceptions of program elements, and potential improvements.”	Qualitative research. Ethics approval and informed consent obtained. No instrument mentioned for scoring gambling problems	*n* = 103 56% men 44% women Mean age = 43.8 All problem gamblers: Self-excluders (*n* = 53) Not self-excluded (*n* = 50)	“While the program is reaching some of the target group, others are delayed or deterred from entering the program due to low awareness, shame, embarrassment, the need to exclude individually from venues, lack of privacy and confidentiality, and low confidence in venue monitoring.”
Hing et al. (2015) [28]	Maintaining and losing control during Internet gambling: A qualitative study of gamblers’ experiences	Australia	Original research	“This paper provides an in-depth exploration of the psycho-social factors and processes related to maintaining and losing control during Internet gambling. It explores features of Internet gambling leading to loss of control, control strategies used by Internet gamblers, and perceived utility of online responsible gambling measures.”	Qualitative research, Ethics approval and informed consent obtained. Instrument used: Problem Gambling Severity Index (PGSI) [29]	*n* = 25 100% men 0% women Mean age = 39.9 All participants scoring 3+ on the PGSI	“The most frequently identified aspects of Internet gambling leading to impaired control were use of digital money, access to credit, lack of scrutiny and ready accessibility. Participants used a range of self-limiting strategies with variable success. Most considered that more comprehensive responsible gambling measures are required of Internet gambling operators.”
Pickering et al. (2022) [30]	Online self-exclusion from multiple gambling venues: Stakeholder co-design of a usable and acceptable self-directed website	Australia	Online demographics and screening questionnaire. Semi-structured focus groups and interviews	“(1) To elicit key stakeholders’ ideal expectation of a self-exclusion website in terms of its design features and functioning; (2) to identify practical issues that could potentially impact the website development and implementation.”	Qualitative research. Ethics approval and informed consent obtained. No instrument mentioned for scoring gambling problems	*n* = 25 48% men 52% women Mean age = 37.7 5 self-excluders 20 “professional participants”	“Stakeholder perspectives were consistent with content analysis indicating the importance of website user-friendliness, flexibility, supportiveness, and trustworthiness. Participants believed that the entire self-exclusion process should be conducted online, including identity verification, whilst expecting high-level data security measures to protect their personal privacy.”
Pickering et al. (2019) [31]	Consumer Perspectives of a Multi-Venue Gambling Self- Exclusion Program: A Qualitative Process Analysis	Australia	Original research	“Participants were asked open-ended questions about their experiences and opinions of [a multi-venue self-exclusion program for land-based gaming machine venues], including its strengths and weaknesses, and suggested improvements for future consumers.”	Qualitative research. Ethics approval and informed consent obtained. Instrument used: Problem Gambling Severity Index (PGSI) [29]	*n* = 20 55% men 45% women Mean age = 46.2 13 current self-excluders 7 former self-excluders	“Participants lacked confidence in venues’ willingness and ability to identify non-compliant gamblers and high- lighted the need for vastly improved detection systems. The quality of interactions with venue staff in relation to self-exclusion were mixed; counsellor support, however, was perceived as important from beginning to end of a self-exclusion period.”
Tong et al. (2018) [32]	Public Awareness and Practice of Responsible Gambling in Macao	China (Macau)	Original research	“To explore means for enhancing the responsible gambling (RG) campaign, we studied Macao residents’ interpretation and adoption of RG practices. In Study 1, a random community sample was collected to assess the extent to which common RG practices were adopted. In Study 2, focus group discussions were conducted to explore how RG was conceptualized.”	Qualitative research. Ethics approval and informed consent obtained. Instruments used: DSM-5 criteria for gambling disorder [33] and the evaluation items of the Responsible Gambling Organizing Committee [34]	Study 1 *n* = 1020 45% men 55% women Mean age = 44.49 Random community sample Study 2 *n* = 25 24% men 76% women Age range: 21–63 Non-problem gambling disorder gamblers	“We found that people in Macao may not conceptualize RG in the same way as the government envisions it, and it may partially be a result of their limited knowledge and a lack of confidence in the stakeholders, such as the gaming operators. Our participants also displayed low trust toward counseling service institutes, which may be a result of the non-transparent procedure involved in help-seeking.”

## Data Availability

All data are available from the corresponding author upon reasonable request.

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
