# Peer review of "Qualitative Experience of Self-Exclusion Programs: A Scoping Review"

_ijerph, 2023, doi:10.3390/ijerph20053987_

Round 1

Reviewer 1 Report

This review examines the research on self-exclusion programs as a strategy for responsible gambling for people with gambling disorders. From electronic databases the authors performed a scoping evaluation of papers on the subject with a final set of 6 papers chosen for the review. Overall, the analysis indicates that self-exclusion programs have the potential to be useful tools for people with gambling disorders, but additional research is needed.

Comments bellow on each section

Introduction

The beginning of this article clearly identifies the issue of gambling disorder as a significant public health concern and cites pertinent research on the topic. However, more specific information and illustrations of self-exclusion programs would be helpful in the introduction. The authors refer to the fact that different jurisdictions have different self-exclusion policies and that some have harsh penalties for violators, but they don't provide any specific instances of these policies or programs. The authors also mention research that attempted to assess the efficacy of self-exclusion programs, although they don't give specific instances of these studies.

The introduction gives a clear and succinct explanation of the issue of gambling addiction overall, but it could use additional detail and specificity in regard to self-exclusion programs and examples. This would make the article more substantial and illuminating by helping to better link the goals and scope of the current study with the existing body of research.

Materials and Methods

This scientific paper's material and methods part is effectively written and organized in accordance with PRIMA standards (Preferred Reporting Items for Systematic Reviews and Meta-Analyses). It gives a precise and in-depth explanation of the study's data collection procedures, analysis methods, and research strategy. The steps used to guarantee the accuracy and reliability of the data acquired have been clearly laid forth by the authors.

The paper, however, might benefit from having more thorough details about the screening procedure. It would be helpful to have more details on the specific people participating in the process, such as the number of reviewers and the standards used to determine study eligibility, even though the techniques used to find pertinent studies are given. Additionally, more information on the screening procedure would improve the study's transparency and reproducibility. This is crucial to defending the choice of publications and their inclusion in the systematic review. The material and methods part are well written and arranged overall, although the work would benefit from further details on the screening procedure.

Results/Discussion and Conclusion

This scientific paper's results, discussion, and conclusion sections are well-written and give a concise but compelling summary of the study's findings. The authors were able to relate the research objectives with the results, they also acknowledged limitations.

Author Response

Introduction 

The beginning of this article clearly identifies the issue of gambling disorder as a significant public health concern and cites pertinent research on the topic. However, more specific information and illustrations of self-exclusion programs would be helpful in the introduction.  

Answer: We have added a sentence to give a common concrete example of self-exclusion. Thank you for your comment, we agree this clarification makes it easier to grasp the self-exclusion concept as it unfolds in a real-life setting. 

The authors refer to the fact that different jurisdictions have different self-exclusion policies and that some have harsh penalties for violators, but they don't provide any specific instances of these policies or programs.  

Answer: Thank you for this good point. We have added a concrete example of New Zealand legislative measures when a self-exclusion order is breached. 

The authors also mention research that attempted to assess the efficacy of self-exclusion programs, although they don't give specific instances of these studies. 

Answer: We have added a concrete example to illustrate this point. 

The introduction gives a clear and succinct explanation of the issue of gambling addiction overall, but it could use additional detail and specificity regarding self-exclusion programs and examples. This would make the article more substantial and illuminating by helping to better link the goals and scope of the current study with the existing body of research. 

Answer:  Thank you for this comment. We agree that examples can help the readers. We believe that the examples provided in previous comments (1-3) will better inform the reader. Thank you for your suggestions. Hopefully, this helps readers understand how self-exclusion programs can play out and be applied empirically. 

  

Materials and Methods  

This scientific paper's material and methods part is effectively written and organized in accordance with PRIMA standards (Preferred Reporting Items for Systematic Reviews and Meta-Analyses). It gives a precise and in-depth explanation of the study's data collection procedures, analysis methods, and research strategy. The steps used to guarantee the accuracy and reliability of the data acquired have been clearly laid forth by the authors. 

Answer: Thank you for this comment. 

The paper, however, might benefit from having more thorough details about the screening procedure. It would be helpful to have more details on the specific people participating in the process, such as the number of reviewers and the standards used to determine study eligibility, even though the techniques used to find pertinent studies are given. Additionally, more information on the screening procedure would improve the study's transparency and reproducibility. This is crucial to defending the choice of publications and their inclusion in the systematic review.  

Answer: The specific team members participating in the screening process, as well as the eligibility criteria, can be found in Section 2.2 “Study Selection and Data Extraction” (above and below Figure 2). We have revised this section and hope this new version is clearer. We fully agree on the importance of full disclosure for transparency and reproducibility, as these are essential elements of research ethics. Thank you for pointing it out. 

The material and methods part are well written and arranged overall, although the work would benefit from further details on the screening procedure. 

Answer: Please refer to the above answer on the details of the screening procedure.  

Results/Discussion and Conclusion 

This scientific paper's results, discussion, and conclusion sections are well-written and give a concise but compelling summary of the study's findings. The authors were able to relate the research objectives with the results, they also acknowledged limitations. 

Answer: Thank you! 

Reviewer 2 Report

The authors carried out a scoping review of the qualitative experience of self-exclusion programs aimed at people with gambling disorders. They searched for articles in 10 databases, finally including 6 articles after a selection process. They found that these programs are generally seen to be an effective responsible gambling strategy, however, there are some aspects that need to be improved.

My comments are:

In figure 2: “Full-text references excluded (not primarily about the perspective of stakeholders on self-exclusion)) (n = 20)”. Verify and correct, if applicable.

Mention whether the studies of the included articles have informed consent and approval by an ethics committee, if applicable.

Mention the instruments applied in the studies, if any.

It is suggested to show the results summarized in a table, focusing mainly on the qualitative experiences, since it is the key theme of the review.

Author Response

In figure 2: “Full-text references excluded (not primarily about the perspective of stakeholders on self-exclusion)) (n = 20)”. Verify and correct, if applicable. 

Answer: This was an error. Thank you for pointing it out. We have corrected the number to n=18. 

Mention whether the studies of the included articles have informed consent and approval by an ethics committee, if applicable. 

Answer: This information has been added to the table. 

Mention the instruments applied in the studies, if any. 

Answer: This information has been added to the table. 

It is suggested to show the results summarized in a table, focusing mainly on the qualitative experiences, since it is the key theme of the review. 

Answer: Thank you for this suggestion. We have decided to keep the current format so that the reader can also understand the overall objectives and results of the studies. We believe this gives interesting context to readers and a more coherent overview of the nature of each study conducted. On the other hand, the result section of our article focuses on the data related to qualitative experiences.